# A System for Autonomous Seaweed Farm Inspection with an Underwater Robot

**DOI:** 10.3390/s22135064

**Published:** 2022-07-05

**Authors:** Ivan Stenius, John Folkesson, Sriharsha Bhat, Christopher Iliffe Sprague, Li Ling, Özer Özkahraman, Nils Bore, Zheng Cong, Josefine Severholt, Carl Ljung, Anna Arnwald, Ignacio Torroba, Fredrik Gröndahl, Jean-Baptiste Thomas

**Affiliations:** 1KTH—Royal Institute of Technology, SCI School, 100 44 Stockholm, Sweden; svbhat@kth.se (S.B.); jsev@kth.se (J.S.); carlju@kth.se (C.L.); aarnwald@kth.se (A.A.); 2KTH—Royal Institute of Technology, EECS School, 100 44 Stockholm, Sweden; sprague@kth.se (C.I.S.); liling@kth.se (L.L.); ozero@kth.se (Ö.Ö.); nbore@kth.se (N.B.); zhcon@kth.se (Z.C.); torroba@kth.se (I.T.); 3KTH—Royal Institute of Technology, ABE School, 100 44 Stockholm, Sweden; fredrik.grondahl@abe.kth.se (F.G.); jean-baptiste.thomas@abe.kth.se (J.-B.T.)

**Keywords:** seaweed farm, algae farm, behavior trees, simulation, mission planning, field testing, system integration, AUV

## Abstract

This paper outlines challenges and opportunities in operating underwater robots (so-called AUVs) on a seaweed farm. The need is driven by an emerging aquaculture industry on the Swedish west coast where large-scale seaweed farms are being developed. In this paper, the operational challenges are described and key technologies in using autonomous systems as a core part of the operation are developed and demonstrated. The paper presents a system and methods for operating an AUV in the seaweed farm, including initial localization of the farm based on a prior estimate and dead-reckoning navigation, and the subsequent scanning of the entire farm. Critical data from sidescan sonars for algorithm development are collected from real environments at a test site in the ocean, and the results are demonstrated in a simulated seaweed farm setup.

## 1. Introduction

We will outline the challenges and opportunities in operating autonomous underwater vehicles (AUVs) on seaweed farms. We propose a system for using AUVs for inspection. We have built a prototype of the AUV. We have tested parts of the system in simulation and using real data collected from the AUV in a seaweed farm. The motivation for using AUVs in this application is mainly one of reducing cost and risk. At the same time, seaweed farming promises to be very profitable while it helps to feed the world in a sustainable way.

According to the UN’s medium-variant projection, the human population will likely increase from 7.7 billion people in 2019 to 9.7 billion by 2050 and reach 10.9 billion by 2100 [1]. The oceans can help to satisfy the global demand for food both directly by sustainable food production or indirectly by the harvesting of cultivated and wild biomass for feed. The current and growing population is a challenge connected with hunger, malnutrition, and micronutrient deficiencies, through the growing global demand for food and biomass.

Compared to 2012, the total food, feed, and biofuel demand is projected to increase by 40–54% by 2050 [2]. Our oceans are home to a large number of resources that are currently marginally exploited and that may improve food security and wellbeing for humans. To sustainably unlock the potential of our oceans to support society with healthy and low-carbon food, focus is needed on the aquaculture and mariculture of low-trophic species [3], such as microalgae, macroalgae (seaweeds) and shellfish, which could result in a 50 to 100 fold increase in productivity in comparison with today’s cultivation of carnivorous fish such as salmon [4]. When taking a life cycle perspective on the environmental burdens of production of low-trophic species, studies have also shown the use of nutrients, carbon emissions, and freshwater to be low, even more notably when compared to other blue foods [5] and animal-based sources of protein [6]. When these low-trophic species are harvested, nutrients and carbon are moved from sea to land, closing the loop notably on finite resources such as phosphorus [7]. Low-trophic species thus also may hold the key to mitigating eutrophication, at least at a local level. Depending on the use of the biomass, for example for long-lived biomaterials or as a fertilizer, the CO_2_ may be captured for a longer period of time and also work as a temporary carbon sink [8]. Furthermore, studies of large seaweed cultivation in Sweden have also shown a positive effect on biodiversity in and around the cultivation sites [9], so regular inspections could help to document local environmental change.

The sustainable future is “Blue” and we need to unlock the potential of the ocean for energy systems e.g., wind and wave energy as well as increase aquaculture in order to achieve a prosperous future for humanity. At the same time, the ocean environment is a harsh and demanding environment to work in and very expensive ships and diving is needed. Large-scale macroalgae cultivation can benefit from the automation of the inspection task. In particular, replacing current divers and data collection from boats with fully automated inspection systems using AUVs could substantially reduce risks and costs [10]. Underwater robots can support licensing and water quality measurements, facilitate infrastructure inspections, and monitor the state of the crop. AUVs have for instance been used to monitor seaweed farms in recent work such as [11,12]. Moreover, the current boat-centered methods consume fossil fuels, which contributes to pollution and global warming [13]. Using AUVs can drastically reduce the carbon footprint of these operations and as a result, as these farms are scaled up, automation becomes more and more cost-effective.

Operations with AUVs at sea, however, pose their own challenges regarding autonomous vehicle navigation, perception, control, and planning. This is due to the limited sensing capabilities available underwater, where electromagnetic-based sensors are quickly rendered ineffective, together with the unpredictable dynamics of the open-water domain. Some of these challenges can be overcome with high-grade navigation solutions and sensors. However, for the automation of the inspection tasks to be cost-effective at a large scale, an expensive AUV with powerful thrust and high-end navigation and sensing is not attractive. We propose a system centered around an affordable, small AUV with a compact sensor suite. Such a system then implies increased challenges for navigation especially when underwater with no GPS and a dynamically varying structure like the seaweed farm. The reduction in size also entails a more careful design of the electrical layout of the vehicle in order to avoid interference, since the various sensors, motors, and computers need to be placed in near proximity to one another. Regarding the data collection, cameras are essential to provide information about the state of the crops. Although cameras can also be used for autonomous navigation in aquaculture farms as in [14], sidescan sonar could be better suited as it can give a range and bearing to the ropes and buoys from a further distance regardless of the water conditions. Cameras on the other hand can not see very far in water and do not directly give the range to the lines, something that is important as it is the control parameter for the inspection task. Smaller AUVs do come with other advantages besides cost as compared to larger ones. They can be launched by hand and can more easily maneuver in tight spaces. The lower thrust and simpler navigation sensing mean that the AUV will not be able to control drift in the dead-reckoning estimates while underwater. In a seaweed farm, however, it is more relevant to navigate relative to models of the farm built up from relative measurements from the AUV such as the sonars and cameras as opposed to any absolute geo-positioning of the farm [15].

In this paper, we describe a new generation of AUVs that could be used as inspection robots in the coming blue revolution and particularly for the new generation of sea farmers in the expanding field of kelp (brown macroalgae, specifically *Saccharina latissima*) cultivation. We propose a system for kelp farm inspection including bespoke components such as the vehicle and sensors, as well as the software parts. These include a graphical user interface (GUI) for planning an inspection mission, and a behavior tree (BT) framework for executing inspection plans, control, sensing, and navigation solutions. This paper is organized as follows: in Section 2 we describe the kelp farm and the inspection requirements, Section 3 is the robot description, Section 4 is an overview of the planning, Section 5 is on detection of the rope and buoy, Section 6 is on the initialization, Section 7 is on the line inspection, Section 8 covers the simulations Section 9 is a discussion, and Section 10 is the conclusion.

## 2. Seaweed Farm Description and Inspection Requirements

A typical kelp farm consists of a line of buoys with the ends moored to the bottom (see illustration in Figure 1). The kelp grows down from the rope connecting them about a meter or two below the surface. These lines can be hundreds of meters long and the farm is laid out as a number of these lines in parallel several meters apart (see example site from the Swedish west coast in Figure 2. In the northern hemisphere, ropes seeded with juvenile kelp are typically deployed in the fall and harvested in late spring or early summer, depending on the locality. By that time the algae has grown to several meters in length. During this growing period storms, boats, or other causes can disrupt and displace the farm. For this reason and to monitor the overall health and growth stage of the kelp the farm needs periodic inspection under the surface. The more frequently these occur, the sooner problems can be addressed, crop growth can be predicted, and harvesting can be planned for maximizing production and profits.

## 3. Robot Description

The AUV SAM [16,17] is used to demonstrate the methodologies for the seaweed farm inspection. SAM is a torpedo-shaped, underactuated AUV whose unique actuator configuration makes it highly maneuverable and hydrobatic, Figure 3. The actuator subsystems include counter-rotating propellers for propulsion and a thrust vectoring nozzle for maneuvering. A variable buoyancy system facilitates static depth control by pumping water in and out of a tank. A movable battery pack enables static pitch control through longitudinal changes to the center of gravity (c.g.) position while rotating counterweights enable transversal c.g. changes for static roll control.

With this actuator configuration, both static and dynamic changes to state are possible, and advanced controller policies (such as model predictive control, MPC) offer unique opportunities for steering the vehicle. Navigation and payload sensors include an inertial measurement unit (IMU), a compass, a GPS (for surface operations), a Doppler velocity logger, cameras, sidescan sonar, and environmental sensors. The AUV is designed to be very slender with no external antennas, fins, or rudders that can get entangled in cluttered spaces. SAM uses these unique maneuvering features to enable operations in the confined environments of the seaweed farm.

Software-wise, the autonomy stack on SAM runs on the robot operating system (ROS) environment and its sub-components:**Mission planning through Neptus:** A graphical interface for the operator to plan out the mission on a world map and monitor the current status of the vehicle while it works.**Robust mission execution using behavior trees:** A behavior tree ensures the safety and compliance requirements while the mission executes, disallowing unsafe behaviors autonomously.**Path planning using spline-fitting:** Given a set of waypoints as the mission, a spline that is tuned to the vehicle is fit on the waypoints to ensure that the vehicle can reach each waypoint.**Feedback control using cascaded PID controllers:** PID controllers that generate the required signals for the actuators are controlled by other PID controllers that generate signals for the desired actuator set-points which helps keep the actuators from getting damaged while keeping control of the vehicle.**Dead reckoning with an extended Kalman filter:** An extended Kalman filter is used to fuse together the external and inertial measurements the vehicle collects to estimate the current position and orientation of the vehicle when GPS is not available underwater.

All the software subsystems within ROS can be validated using the Stonefish simulator environment [18], thus enabling rehearsals of mission operations prior to deployment. For an overview of the general system, see Figure 4.

## 4. Overview of the Autonomous Inspection Planning

The proposed system is designed to allow the operator to quickly set up an inspection plan with only partial information about the seaweed farm. Then she would launch the AUV and have it find its way to the farm, carry out the inspection and return with the data. The steps of a typical plan set up by the operator prior to launch are:Vehicle is launched, and a GPS fix is acquired.The AUV mission is started and the vehicle follows a plan to reach the farm’s vicinity.The AUV circles around the farm, using sidescan sonar to detect buoys and construct a relative-to-AUV map of the farm.Move to the beginning of the seaweed line, using the constructed map.Start surveying the line according to the previously constructed map.Transition to line-following once some of the lines have been detected using the sidescan sonar.At end of the line execute a 180 turn to position the AUV on the other side of the line (between two lines now).Follow the line using the previous detections of the first line seen from the first side and known prior line spacing until sufficient detections of the next line are found.Repeat from (6) until complete.Once all lines are surveyed, finish the planned mission and return to base.

Mission planning is done using the Neptus (https://github.com/LSTS/neptus (accessed on 27 June 2022)) open source software package from Laboratório de Sistemas e Tecnologia Subaquática (LSTS) (https://lsts.fe.up.pt/ (accessed on 27 June 2022)), which has been integrated into the ROS ecosystem. The integration of Neptus into ROS is accomplished through a message bridge that can do two-way translations of intermodule communication (IMC) (https://github.com/LSTS/imc (accessed on 27 June 2022)) messages between Neptus and ROS. The Neptus GUI interfaces with the behavior tree (BT) to set up and monitor the mission in a user-friendly manner. The mission is planned on Neptus as a list of waypoints, where each waypoint contains specific requirements like depth, speed, and which sensors enable. The BT then receives this plan and uses Neptus to display the progress, current maneuvers, and any other feedback relating to the mission being executed. When a communication link is established, an operator can use Neptus to stop, pause, replace, and resume a running mission plan.

A BT is a reactive decision-making structure that is comprised of Sequences, Fallbacks, Actions, and Conditions. Each child node can return either Running, Success, or Failure to its parent. Actions and Conditions are allowed only as leaf nodes while Sequences and Fallbacks are allowed as only inner nodes. The BT operates at a fixed frequency, sending a so-called “tick” down from the root to its children. Once a “tick” is received by a leaf node, it is executed, and it returns immediately. The next child is ticked if the parent is a Sequence and it received a Success or the parent is a Fallback and it received a Failure. If a leaf node is not ticked, it does not execute or stops its execution if it was running previously. If a child returns Running, it is repeated until the root without ticking any sibling nodes along the way. The Running return state indicates that the node will take more than one tick’s time to finish and decide its return, which usually indicates a long-running action such as moving to a waypoint. Such a structure allows the designer to prioritize actions, sequence missions, and generally make sure the vehicle operates safely while keeping the overall setup modular and easy to understand [19,20]. See Figure 5 for a condensed view of the BT used on SAM.

For AUVs, there are two categories of safety checks: data integrity and operational limits. The data collection sub-tree of the BT makes sure that the sensors are working and are reading data at the required frequencies. If the vehicle cannot sense its environment and the data flow is interrupted, an emergency is triggered. The safety sub-tree handles the checks for operational depth, altitude, and leaks so that the vehicle stays within its operational bounds, regardless of the running actions. In the case of the vehicle exceeding its operational limits or in the case of a leak, the BT runs an emergency surfacing action while stopping everything else the vehicle might be doing. With these sub-trees acting as guards, the vehicle is allowed to run autonomous missions without worrying about safety or data integrity.

Under normal conditions, without a given mission, the BT will wait for the operator. As long as all the previous conditions are met, the mission is allowed to run once the operator gives the signal. For example in Figure 5 the autonomous line-following behavior has higher priority than the waypoints given by the operator. If there are no lines to follow, the vehicle will fall back to following the operator’s plan. If at any point data integrity is broken or the operational limits are exceeded, both line following and waypoints are ignored and emergency action is executed until the vehicle is safe again.

## 5. Seaweed Farm Rope and Buoy Detection

Preliminary field tests were carried out at the marine field station Kristineberg on the Swedish west coast, Figure 6, where the Swedish Maritime Robotics Centre (SMaRC (https://smarc.se (accessed on 27 June 2022))) has a small seaweed test site. The testbed is equivalent to the seaweed farm built up in the simulator described in this paper and consists of two lines of seaweed ropes attached to supporting buoys at the surface and anchored to the sea bed by concrete blocks. The farm is approximately 15 by 15 m and the ropes are situated at about 2 m depth. Here we have been able to test the rope detection scenario in a real setting so that the algorithms we are developing are trained and tested on real data. Data have been collected during a number of visits to the farm in different conditions during the year April, June, and November).

The relative navigation in the seaweed farm with respect to the ropes is a critical step in the operation. Direct detection of the rows of seaweed to be inspected is important as the AUV navigation while underwater is subject to substantial drift and the layout of the farm is subject to change over time. Thus, detection is needed for both finding the farm and for control along the rows.

Although cameras will be needed to provide the actual inspection data on the seaweed condition, the need to potentially look for the farm using sensor information renders underwater cameras less useful in this scenario, as cameras can only obtain interpretable images at short range, given clear water conditions and sufficient lights. Sidescan sonar, an acoustic sensor that provides long-range observations without constraints on the water quality, is instead employed for control feedback sensing. Apart from being able to provide high-resolution observations of ropes and buoys, sidescan sonar has been shown to provide useful signals for seaweed growth rate, making it an ideal sensor for seaweed farm monitoring [21]. An example of a 2D sidescan sonar “image” of a segment of the inspected seaweed farm is shown in Figure 7. The 2D image is composed of 1D sonar pings stacked vertically.

The presence of ropes and buoys are seen as change points in the 1D sidescan signal, and a window-sliding segmentation algorithm is used for change point detection. A window of fixed size is sliced across a 1D sidescan signal. Within that window, a mean and the variance from the mean are computed. The difference in variance between two adjacent windows is recorded as the score [22]. The higher the score, the greater the change in signal strength. This algorithm is not significantly costly in terms of computational complexity, being of the order of the length of the sonar return vector.

More specifically, let *s* denote the 1D sidescan signal with total length of *T* and $s_{i:i+t}$ denote a window of the signal of length *t* starting at index *i*. For each window, a least square deviation from the window mean can be computed as follows:$$c\left(s_{i:i+t}\right)=\sum_{n=i}^{i+t}\left|\right|y_{n}-\bar{y}{\left|\right|}_{2}^{2}$$
where $\bar{y}$ refers to the mean of the signal strength within the window, i.e., $\bar{y}=\frac{\sum s_{i:i+t}}{t}$.

The discrepancy between two adjacent windows can be computed using the following formula:$$d\left(s_{i:i+t},s_{i+t:i+2t}\right)=c\left(s_{i:i+2t}\right)-c\left(s_{i:i+t}\right)-c\left(s_{i+t:i+2t}\right)$$

This discrepancy indicates the cost gained by splitting the sub-signal $s_{\left(i:i+2t\right)}$ at index (i+t). To detect change points in a signal, a window of size *t* is slid across the signal, and the index with the highest discrepancy is selected as the change point.

The nadir is the first bottom-hitting acoustic return received by the sidescan. By definition, the change in signal strength is likely to be high. To avoid misdetecting nadir as objects, the detection algorithm runs two window-sliding sessions, first identifying the nadir and then identifying any objects in the water column. The same algorithm can be used to detect both ropes and buoys by changing the sliding window size and signal change ratio. An example rope detection procedure is shown in Figure 7. At the top the raw sidescan sonar ping with rope signal is shown, time corresponds to rows and increases going down in the image. The middle graph shows nadir segmentation results. The blue segment contains a signal from the water column (before the first bottom return) and is used for rope and buoy detection. The bottom graph shows the rope segmentation results. The change point represents the location of the rope detection. The detection, together with the confidence score, is published to ROS and can be subscribed by other ROS nodes. Due to the low time and memory complexity of the algorithm, the change point detection can run in real-time without impacting the vehicle’s power consumption or performance.

## 6. Initialization of the Inspection Plan

Before the actual inspection can begin, the AUV must localize the farm relative to its position and determine the overall layout of the farm, that is, the position of the outer lines and buoys. It is only expected that a rough layout and position of the buoys on the farm are known at the start of a mission. This is very much in line with a real scenario where the buoy locations may not be exactly known in the first place and may also have drifted during e.g., rough weather conditions. The AUV begins by navigating on the surface to the approximate farm location given by GPS coordinates. It then circumnavigates the farm according to its navigation and prior information on the farm layout. During this phase, the AUV stays relatively far from the farm and attempts to detect the buoys in its vicinity.

To handle potential misdetections of buoys, the raw detection results described in Section 5 are fed into a variational Gaussian mixture model (VGMM). The VGMM is instantiated with a maximum amount of classes equal to the number of buoys in the farm, allowing it to cluster buoy detections and be robust against outliers. Once the buoys have been detected sufficiently often to give reasonable confidence in their relative locations, we do a piece-wise linear fit through the buoys such that the resulting lines must align with the a priori known farm orientation (compass direction). A depiction of this line fit is given in Figure 8. The results from this VGMM line fit serve as an updated seaweed farm map and are used to position the AUV for more close-up seaweed row inspections.

## 7. Line Following Inspection

The ropes that the seaweed hangs from do not lie perfectly straight. In order to hold the AUV at the proper distance for visual inspection, a line following control behavior was used. The idea is to use the last few detections of the rope to project a straight line and then to control the vehicle to a path offset from that by an appropriate amount. This is similar to a wall following the behavior of an indoor robot navigating a corridor. After moving along the initial route long enough to collect some detections of the rope, a random sample consensus (RANSAC) regressor [23] is used to find the inliers and fit a straight line to them, Figure 9.

In order to follow the straight line path effectively, a line-of-sight guidance law based on look-ahead steering (see [24]) is used to provide a relevant heading set point to the onboard flight controllers. The heading set point is computed as
(1)$$\psi_{d}=\psi_{p}+\psi_{r},$$
where $\psi_{p}$ is the angle tangential to the straight line path while $\psi_{r}$ is the velocity-path relative angle. The path-tangential angle can be calculated from the intercepts as
(2)$$\psi_{p}=\arctan\left(\frac{\Delta y}{\Delta x}\right),$$
while the velocity-path relative angle can be computed as
(3)$$\psi_{r}=\arctan\left(\frac{-e}{\delta }\right)$$
by considering the cross-track error *e* between the current AUV position and the desired path, and ensuring that the AUV intersects with the path within a specified look ahead distance δ (see Figure 10).

## 8. Simulated Field Test Scenario

The test site for the seaweed farm at the marine field station Kristineberg (https://kristinebergcenter.com (accessed on 27 June 2022)) is built up in a mission simulator (https://github.com/smarc-project/smarc_stonefish_sims (accessed on 27 June 2022)) using the open source Stonefish simulation tool [18]. The test site simulation is divided into two parts, one being the numerical model of the bathymetry around the Kristineberg field station and the other being the small seaweed farm model. The seaweeds in the simulation are modeled as warped, elongated rectangles. Figure 11 gives a close-up view of the seaweed farm, as well as the AUV operating underwater during a seaweed row inspection behavior. An integrated view of the two aforementioned pieces (the field station and the seaweed farm), together with the AUV model, is shown in Figure 12.

The entire ROS-based software architecture is integrated within the simulation environment including interfacing with the mission planning software Neptus. The simulation enables validation of the entire software pipeline so that full missions can be rehearsed in a virtual environment. New individual software packages, as well as the entire software system of SAM, are integrated and can be rapidly tested within this setup. Such a testing environment allows for rapid and resource-light development of each piece of the system.

Results of a simulated mission to inspect the algae farm are presented in Figure 13. The BT executes a pre-defined mission plan from Neptus, and a waypoint-following action guides the SAM AUV to go to each waypoint using the line-of-site (LOS) guidance law to follow a straight-line path, followed by the inspection of the algae by going through the lines. This simulation helps validate the control and mission execution pipeline. Some control data from this are shown in Figure 14.

## 9. Discussion

The approach to navigation and control taken in this system is to rely on the recent sidescan sonar measurements of the buoys and lines. This should allow robustness to large errors in the navigation as long as these objects can be brought into view of the sonar. The relatively long range and insensitivity to water conditions of the sonar help in this regard. Additionally, by building relatively simple models such as buoy measurement clustering and line fitting, outliers can be found and ignored.

The work shows promising results in terms of line detection with sidescan sonars enabling operations in a variety of water conditions and at relatively large distances compared to camera-based image recognition. Promising results are also seen from collecting the data with a real AUV equipped with sidescan sonars. It should however be noted that there are still a number of challenges to overcome before these systems are fully operational in the envisioned settings. The main challenges identified are robust detection with varying sizes of the algae (juvenile to fully grown); operations in a tightly spaced farm where multiple lines can be seen on both sides. Further, sensor and in particular sidescan data robustness in a small AUV with many advanced sub-systems operating in a tight encapsulated space is also an additional challenge. Noise and high-frequency disturbances from e.g., motors and servos on board are easily propagated through the powering system of the AUV into sensor data. Finally, as the main sensors on board the AUV for mapping, navigation, and underwater communication are based on acoustics, it is also essential that these systems are robust enough to operate in parallel without interfering with each other.

In order for there to be any benefits from this system for the farm operator there will need to be actual data on the status of the seaweed farm collected by sensors on the vehicle. These would be video but could also include chemical and water quality sensors. Further processing to get numerical assessments of, for example, biomass estimates from the video and sonar are needed but that is not part of our work so far.

Ultimately these systems could be extended with an underwater base into which an AUV could dock for recharging and data communication. Resident AUVs would have a long-term presence and allow near continuous data collections on the condition of the farm. With a robust base system in place, additional functionality can easily be added, e.g., new sensors that could also monitor the water quality.

Beyond the capabilities developed and demonstrated specifically for kelp farms in this paper, such technologies could have numerous other applications. While automated aquaculture monitoring has been proposed in [25,26], these are designed as fixed camera systems and lack the mobility that would be needed to collect information on a seaweed farm spread over a large area. Other types of aquaculture support activities could be performed for other seaweeds, bivalves, fin-fish [14], or other marine species aquaculture, however, the usefulness could be extended to other sectors by developing capabilities such as infrastructure inspection (e.g., underwater power or fiber optic cables), especially at depths where diving is impractical. Furthermore, it is now foreseeable that marine researchers could have smart AUVs at their disposal within a decade, enabling the collection of invaluable data that could complement or outperform human data collection and sampling methods. This has the potential to transform marine science in the same way archaeology was revolutionized with satellite and aerial ground penetrating imaging techniques. This paper presents a demonstration of a specific application of AUV technology, though a world of possibilities will be unlocked in the coming years.

## 10. Conclusions

We have proposed a system for automatic inspection of seaweed farms. A realistic simulation environment has been built and used for the validation of the system. Individual parts have been validated on data collected by the AUV in a seaweed farm. Other parts of the system have been validated at sea.

The system has been realized on a working prototype. The software architecture is tested both in a realistic simulation and on the AUV. The sidescan sonar is used for the feedback control of the inspection action. The detection needed for that control has been tested on real data collected from the AUV in the seaweed farm. The control action has been tested in the simulator.

Future work will include collaborations with local initiatives in Sweden, such as Nordic SeaFarm AB (https://www.nordicseafarm.com (accessed on 27 June 2022)), on the need for underwater robots as essential tools in seaweed farming. The numerical evaluation of the health and growth status of the seaweed farm from sensor data including cameras is also future work.

## Figures and Tables

**Figure 1 sensors-22-05064-f001:**
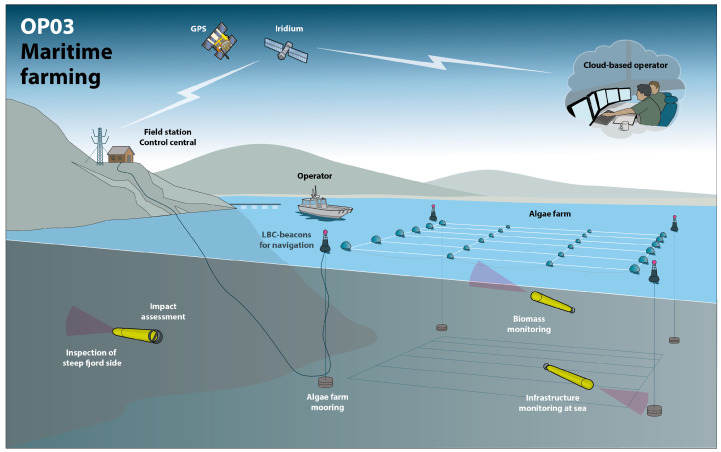
Illustration of scenario with kelp farm and underwater robots (illustration by M. Ek).

**Figure 2 sensors-22-05064-f002:**
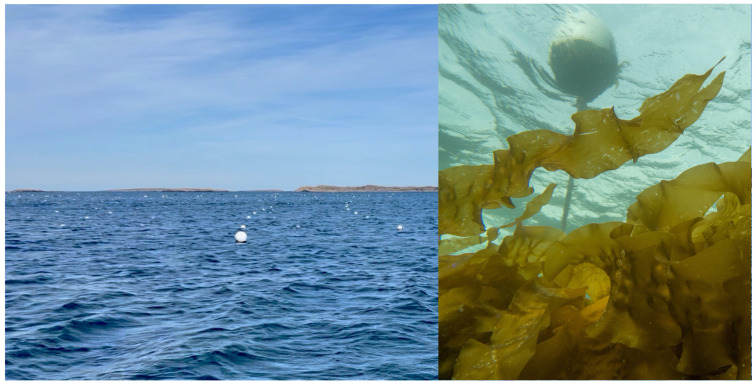
The Nordic SeaFarm site outside of Grebbestad in Sweden. Rows of buoys seen at the surface to the left and the kelp growing on the ropes below the surface to the right. (**Left**: photo courtesy of SMaRC, **Right**: photo courtesy of Nordic SeaFarm).

**Figure 3 sensors-22-05064-f003:**
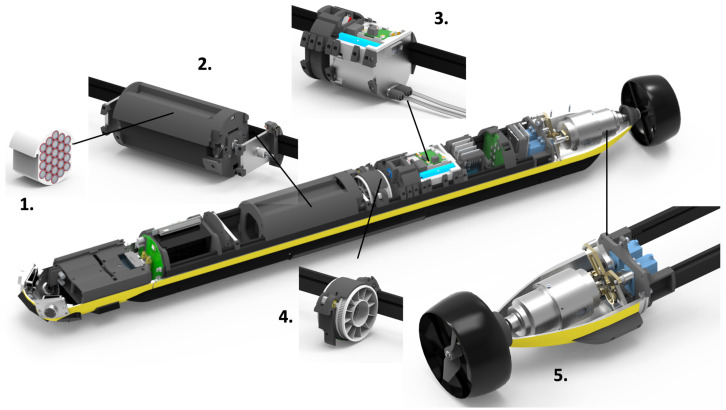
SAM AUV subsystems: 1. battery pack, 2. longitudinal center of gravity trim system (LCG), 3. variable buoyancy system (VBS), 4. transversal center of gravity system (TCG), 5. thrust vectoring system with counter-rotating propellers.

**Figure 4 sensors-22-05064-f004:**
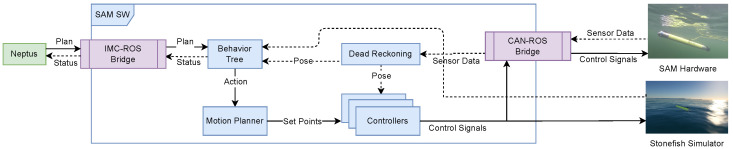
The SAM cyber-physical system architecture integrating a user interface, software, hardware, and simulation tools.

**Figure 5 sensors-22-05064-f005:**
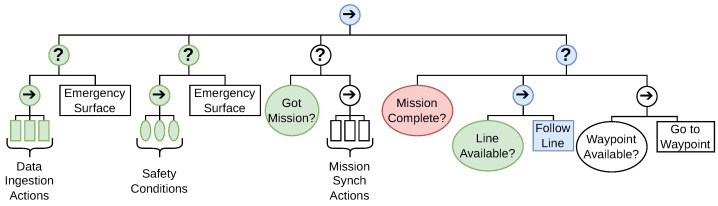
A high-level view of a simple example BT that could be used for the seaweed farm. Inner nodes are Sequences (arrows) and Fallbacks (question marks). Leaf nodes are Actions (rectangle) and Conditions (ellipse). All nodes can return Success (green), Failure (red), and Running (blue). In this example, the vehicle has all the data it needs, is within safety limits, has an incomplete mission, and is currently following a line.

**Figure 6 sensors-22-05064-f006:**
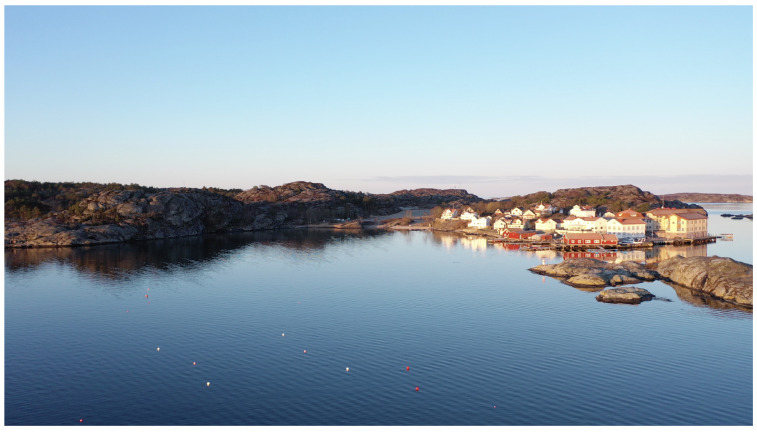
The test site (white buoys in the water) at the Kristineberg marine field station. This is the same site as modeled in the simulator described in this paper (photo: I. Stenius).

**Figure 7 sensors-22-05064-f007:**
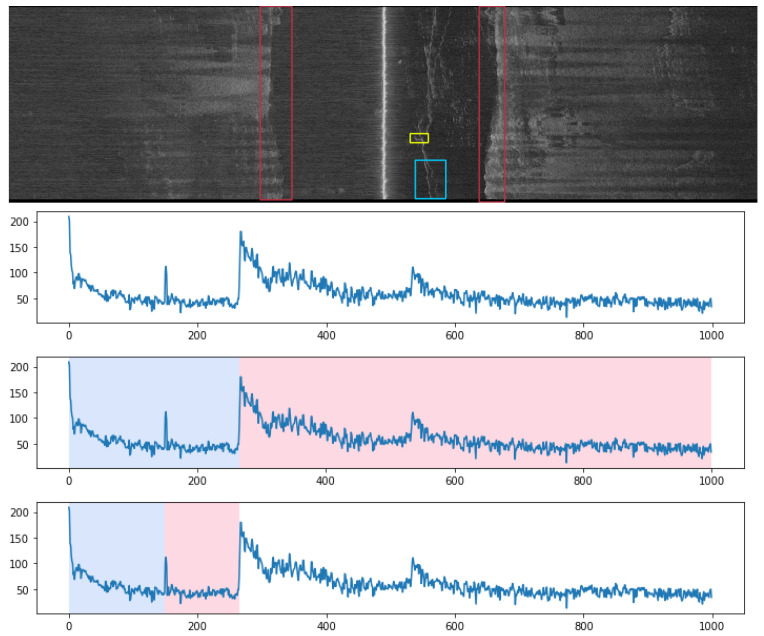
(**Top**): A sidescan sonar “waterfall image” of parts of the seaweed farm. (**Three bottom plots**): The red, blue, and yellow annotations correspond to the nadir, rope, and buoy signals, respectively. The three graphs show the port sidescan sonar signal (a row of the top image) and the corresponding nadir and rope segmentation results.

**Figure 8 sensors-22-05064-f008:**
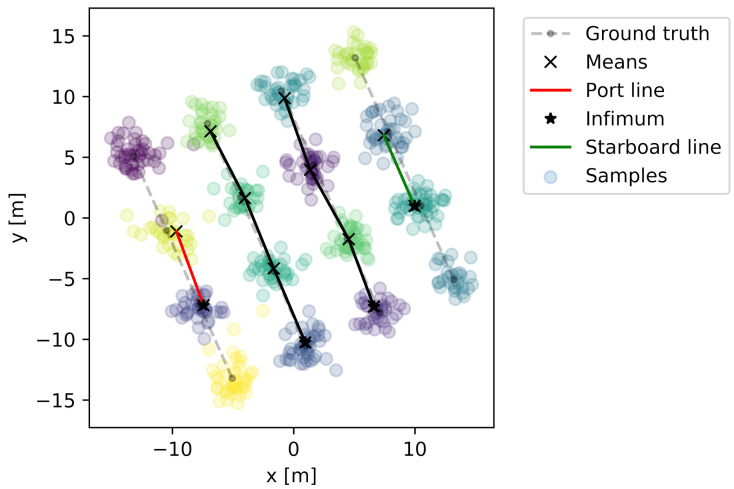
A VGMM fitted to the sensed relative position of the buoys. The different colours indicate the different classes.

**Figure 9 sensors-22-05064-f009:**
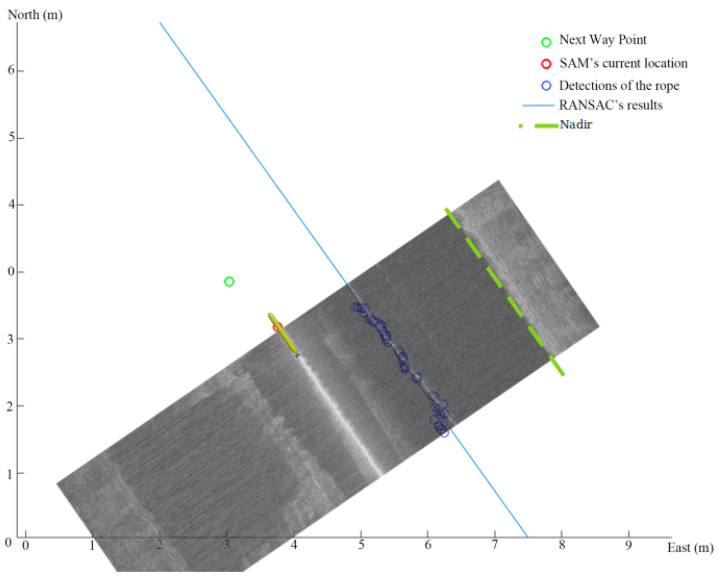
We show how the detections of the rope in the figure are accumulated and fit to a straight line which can then be used to send an offset line to the controller.

**Figure 10 sensors-22-05064-f010:**
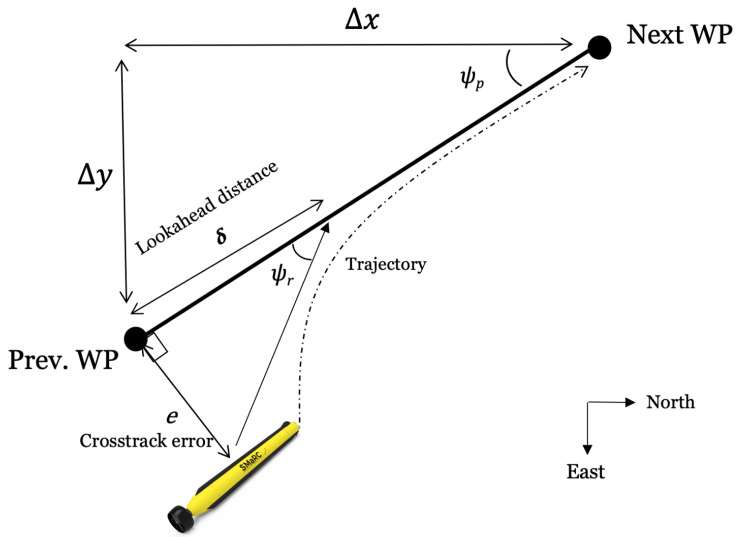
The line-of-sight guidance law considers a lookahead distance δ, the crosstrack error *e*, and the intercepts between waypoints Δy and Δx. This ensures the AUV follows the straight line between waypoints as much as possible by minimizing cross-track error.

**Figure 11 sensors-22-05064-f011:**
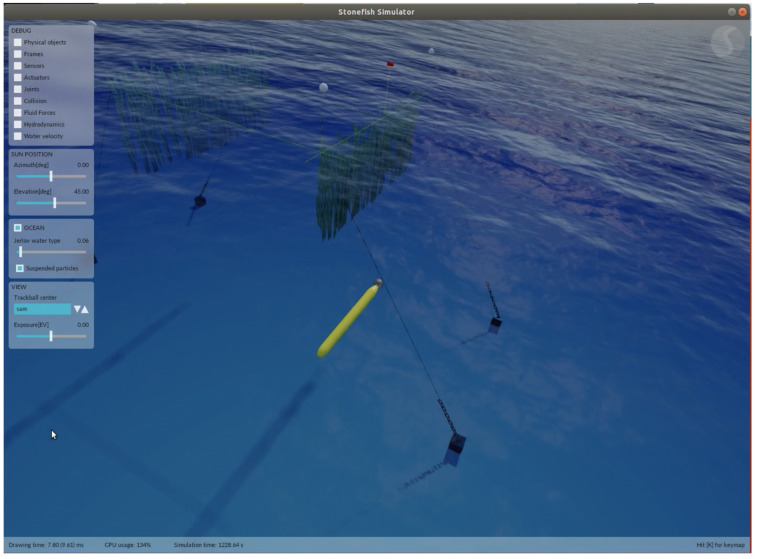
View of the AUV SAM operational in the simulated seaweed farm.

**Figure 12 sensors-22-05064-f012:**
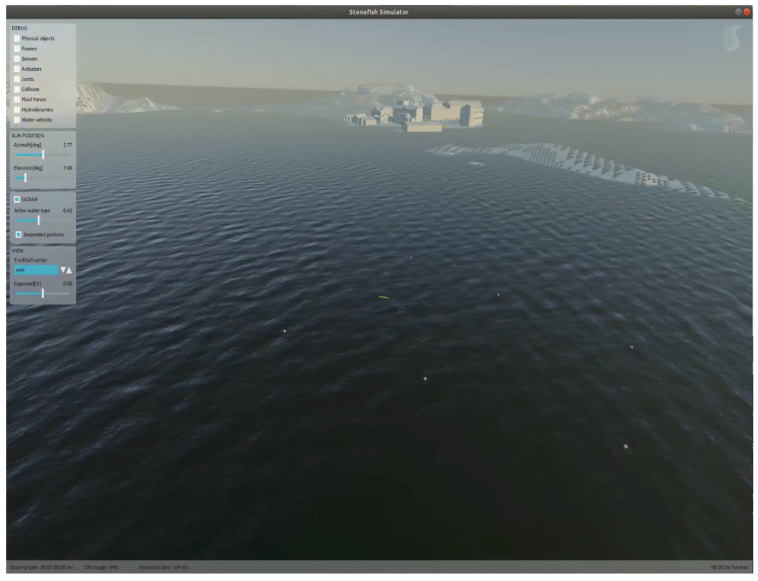
The numerical model (white buoys) seen from the surface of the seaweed farm in the simulator. The SAM AUV (yellow small AUV in the middle of the fig) can also be seen vaguely.

**Figure 13 sensors-22-05064-f013:**
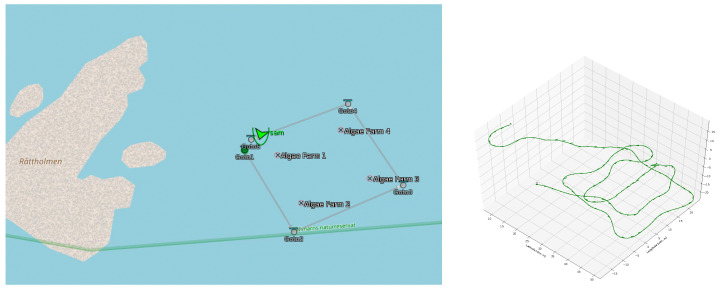
A mission plan to inspect the outside of the seaweed farm in Neptus (**left**). In Stonefish, the AUV follows 4 waypoints in straight lines around the extremities of the farm (shown as crosses in the Neptus figure) at a depth of 2.5 m, and then inspects the inside (**right**). Since this farm has two parallel algae lines, the vehicle does three close-by passes after the initial survey. The vehicle then moves out of the farm.

**Figure 14 sensors-22-05064-f014:**
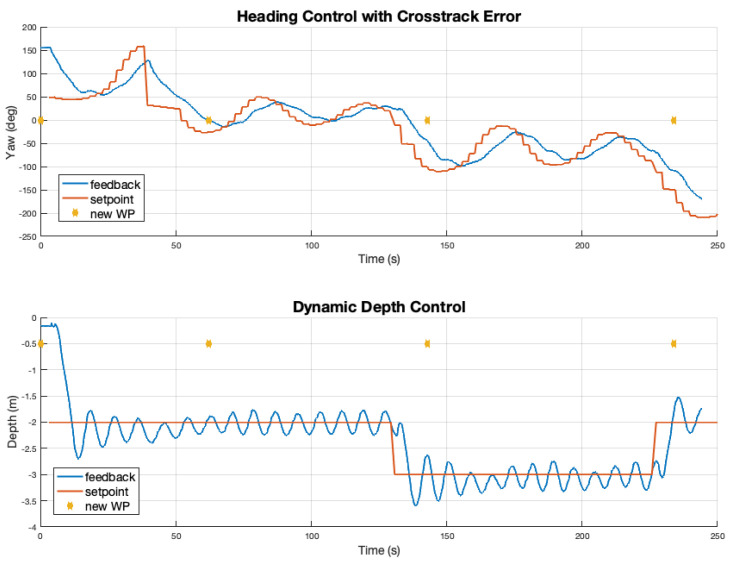
The top plot shows how our PID control follows a yaw command in the simulation while the bottom shows depth control. The orange plots show the set point sent to the actuation controller while the blue is the estimated values used for the feedback to the set-point PID controller. The yaw set-point is computed using the presented LOS guidance law based on cross-track error. The gold stars show points where the way-point from the planner changed.

## Data Availability

Data sharing is not applicable to this article.

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
