# Peer review of "A System for Autonomous Seaweed Farm Inspection with an Underwater Robot"

_sensors, 2022, doi:10.3390/s22135064_

Round 1

Reviewer 1 Report

To the most potential of aquaculture and mariculture of low-trophic species,  a system and methods for operating an AUV in the seaweed farman was proposed to inspect the seaweed farms automaticly.  The workflow  and some challenges of the autonomous inspection seaweed farm mission has been elaborated  in this paper. To be a communication, this paper has been able to illustrate the specific application of AUV technology for the readers commendable. However, the theories used in this paper are mostly obvious arguments fot the researchers, the application scenarios of AUV bring new ideas. There are some comments:

1. If possible, the authors can add some new arithmetic and some data analysis of the realistic simulatioon results which will be attractive the reseachers in the related field.

2. The style of fonts shown in Fig.10 shoud be consistent with the text word.

Reviewer 2 Report

The article describes a method for AUV underwater inspection, which is interesting and promising. However, the scientific language is low.

The article starts with an old prediction that needs an update. I suggest better starting with the scope; What did you do (the aim) and what is your contribution to science. I suggest also a tutorial on "how to write a scientific article."

The figures need a source as per the template. Authors often present assumptions as a fact.

The background of this project should be mentioned. Is it a) fear of world hunger, b ) carbon emissions , or c) profit?

Inconistency of the achronymes, etc.

The methodology and the experimental design should be clearly explained.

The control data is missing!

According to the title, it is not clear, what you want to present to the reader. What is your contribution to science? 

The conclusion is weak and needs improvements.

Good luck.

Reviewer 3 Report

This paper describes a system based on an AUV to inspect seaweed farms automatically. It assesses the quality of the solution through real and simulated cases. The article is well written and clear. The Introduction correctly explains the context and the general goal. Section 2 justifies the seasons for building up the solution. The inspection planning and technical sections are sound.

What I miss in the paper is a clarification of the final scope of the surveys. In particular: Who is the target stakeholder of the collected data? The mentioned detection algorithm identifies objects in a sidescan sonar image, but a farmer would not use this information. Rather (s)he should have a summary or overall assessment of the farm status or an indication of the weak points. Moreover, is there any numerical assessment of the "health and growth stage of the kelp" that comes out of your analysis? Since this point is the declared main target of the monitoring, it should be discussed. 

I report my detailed comments in the following:

Line 13:

"will increase"-> "will likely increase" (indeed you are reporting a theoretical forecast based on complex assumptions)

Line 15:

"direct food production"->"direct sustainable food production"

Line 19:

The citation does not report the FAO technical report name, i.e., it is impossible to verify the reported percentage with the current citation.

Lines 73-75:

Please add a citation here of studies that have compared absolute geo-positioning (top-down) models with adaptive models (bottom-up).

Line 79:

"Saccharina latissima" should be reported in italics because is it a species scientific name.

Lines 83-85:

This part should be organised per section, i.e., by reporting the focus of each section.

Section 3:

Please explain the acronyms (e.g., IMU) when you first introduce them.

Lines 116-122:

This part is useless and unintelligible unless you add one sentence for each sub-component summarising what it does.

Figure 5: The example is clear, but please correct the missing spaces before parentheses.

Line 209:

It is fundamental to describe the window-sliding segmentation algorithm and the detection algorithm. You should also specify that (and if) these algorithms are being executed on board and what is their impact on power consumption. I do not understand if object detection and identification are both performed by the algorithms. Algorithms like these often require powerful hardware and introduce substantial limitations on AUVs. This point should be discussed. Moreover, on board processing hardware power is not specified.

Section 8: 

Is there any numerical assessment of the overall kelp health and growth status emerging from the analysis?

Section 9: 

This section should discuss my point above about the stakeholders and numerical assessment of the kelp health and growth status.

Line 283:

"the" is repeated two times in "the the recent"

Lines 304-305:

"Resident AUVs would have a long-term presence and allow near continuous data collections on the condition of the farm.". This observation is generally valid for all aquafarm types when using intelligent underwater monitoring systems. Examples of intelligent systems for monitoring aquafarms that can sustain your claim here are:

Coro, G., & Walsh, M. B. (2021). An intelligent and cost-effective remote underwater video device for fish size monitoring. Ecological Informatics, 63, 101311.

Almero et al., Development of a Raspberry Pi-based Underwater Camera System for Inland Freshwater Aquaculture. In 2021 IEEE 13th International Conference on Humanoid, Nanotechnology, Information Technology, Communication and Control, Environment, and Management (HNICEM) (pp. 1-6). IEEE.

Livanos et al. Intelligent navigation and control of a prototype autonomous underwater vehicle for automated inspection of aquaculture net pen cages. In 2018 IEEE International Conference on Imaging Systems and Techniques (IST) (pp. 1-6). IEEE.

Please, cite some currently used systems like these to support your sentence.

Round 2

Reviewer 2 Report

Dear Authors,

Thank you for following most of the suggestions.

I recommend accepting the article.

Best regards.